# Evaluation for Potential Drug–Drug Interaction of MT921 Using In Vitro Studies and Physiologically–Based Pharmacokinetic Models

**DOI:** 10.3390/ph14070654

**Published:** 2021-07-07

**Authors:** Hyo-jeong Ryu, Hyun-ki Moon, Junho Lee, Gi-hyeok Yang, Sung-yoon Yang, Hwi-yeol Yun, Jung-woo Chae, Won-ho Kang

**Affiliations:** 1Gwangkyo R&D Center, Medytox Inc., Suwon 16506, Korea; hjryu5767@gmail.com (H.-j.R.); jlee@medytox.com (J.L.); yanggh6940@gmail.com (G.-h.Y.); 2College of Pharmacy, Chungnam National University, Daejeon 34134, Korea; hyeonki@o.cnu.ac.kr (H.-k.M.); 201851000@o.cnu.ac.kr (S.-y.Y.)

**Keywords:** MT921, drug–drug interaction, in vitro studies, transporter, physiologically-based pharmacokinetic model

## Abstract

MT921 is a new injectable drug developed by Medytox Inc. to reduce submental fat. Cholic acid is the active pharmaceutical ingredient, a primary bile acid biosynthesized from cholesterol, endogenously produced by liver in humans and other mammals. Although individuals treated with MT921 could be administered with multiple medications, such as those for hypertension, diabetes, and hyperlipidemia, the pharmacokinetic drug–drug interaction (DDI) has not been investigated yet. Therefore, we studied in vitro against drug-metabolizing enzymes and transporters. Moreover, we predicted the potential DDI between MT921 and drugs for chronic diseases using physiologically-based pharmacokinetic (PBPK) modeling and simulation. The magnitude of DDI was found to be negligible in in vitro inhibition and induction of cytochrome P450s and UDP-glucuronosyltransferases. Organic anion transporting polypeptide (OATP)1B3, organic anion transporter (OAT)3, Na^+^-taurocholate cotransporting polypeptide (NTCP), and apical sodium-dependent bile acid transporter (ASBT) are mainly involved in MT921 transport. Based on the result of in vitro experiments, the PBPK model of MT921 was developed and evaluated by clinical data. Furthermore, the PBPK model of amlodipine was developed and evaluated. PBPK DDI simulation results indicated that the pharmacokinetics of MT921 was not affected by the perpetrator drugs. In conclusion, MT921 could be administered without a DDI risk based on in vitro study and related in silico simulation. Further clinical studies are needed to validate this finding.

## 1. Introduction

MT921 is a new injectable drug developed by Medytox Inc. intended to reduce submental fat, commonly referred to as double chin, through a minimally invasive medical procedure known as injection adipolysis. Its active pharmaceutical ingredient is cholic acid (CA), a primary bile acid synthesized from cholesterol and endogenously produced in the liver of humans and other mammals [1]. In adult humans, the synthesis of CA primarily occurs via the classical pathway of bile acids (BA) biosynthesis [2], inside hepatocytes. It starts with hydroxylation of cholesterol to 7α-hydroxycholesterol by the cholesterol 7α-hydroxylase (CYP7A1) enzyme [3]. Most BAs are reabsorbed in the ileum and sent to the liver by active transport, called enterohepatic circulation. This process involves various transporters, including organic anion transporting polypeptides (OATPs), organic anion transporters (OATs), a Na^+^-taurocholate cotransporting polypeptide (NTCP), an apical sodium-dependent bile acid transporter (ASBT), and a bile salt export pump (BSEP) [4,5,6,7].

CA is an amphipathic molecule with surfactant properties capable of dissolving lipid bilayers. CA is particularly effective in adipolysis, a process that involves destabilizing and disintegrating the lipid bilayer of adipocytes, the major cell found in adipose tissue [8,9,10,11]. A single subcutaneous injection of MT921 into the submental region at strategic intervals results in a significant reduction of submental fat (the data is not shown due to confidentiality issues). Individuals treated with MT921 are likely to have a high body mass index (BMI), so they may presumably be afflicted with some form of metabolic syndrome, such as hypertension, diabetes, and hyperlipidemia, among others [12,13]. Therefore, there is a high probability that these individuals would be on some medication to treat their existing medical condition. Hence, it is imperative to determine the pharmacokinetic drug–drug interaction (DDI) of MT921 to avoid any antagonistic or adverse effect in these patients.

The present study assessed the in vitro pharmacokinetic interaction potential of MT921 following the US Food and Drug Administration (FDA) guidelines [14]. In addition, the effect of MT921 on pharmacokinetics of drugs for metabolic diseases was also investigated by performing physiologically-based pharmacokinetic (PBPK) modeling and simulation studies.

## 2. Results

### 2.1. In Vitro Potential on Transporters

#### 2.1.1. Substrate Specificity of MT921 for OATP1B1, OATP1B3, OAT3, NTCP, and ASBT

Substrate specificity of MT921 for OATP1B1, OATP1B3, OAT3, NTCP, and ASBT was identified in the human embryonic kidney (HEK) 293-OATP1B1, -OATP1B3, -OAT3, -NTCP,–ASBT, and mock cells. To ensure significance, we confirmed the uptake rates of the positive control substrate by the respective transporter-expressing cells and mock cells. The uptake rates of probe substrates (0.022 µM [^3^H]estrone-3-sulfate for OATP1B1 and OAT3, 0.06 µM [^3^H]estradiol-17β-D-glucuronide for OATP1B3, 0.8 µM [^3^H]taurocholate for NTCP and ASBT) in transporter-expressing cells greatly increased compared to those in mock cells and decreased in the presence of representative inhibitors (30 µM cyclosporine A for OATP1B1 and OATP1B3, 50 µM diclofenac for OAT3, 30 µM bromsulphthalein for NTCP, 100 µM deoxycholate for ASBT) (31.3-fold increase in [^3^H]estrone-3-sulfate for OATP1B1; 9.6-fold increase in [^3^H]estradiol-17β-D-glucuronide for OATP1B3; 19.4-fold increase in [^3^H]estrone-3-sulfate for OAT3; 102.1-fold increase in [^3^H]taurocholate for NTCP; 60.3-fold increase in [^3^H]taurocholate for ASBT). Uptake rate was determined at two different concentrations of MT921 (10 µM and 100 µM) in transporter-expressing cells and mock cells. The OATP1B1 transporter showed less than two-fold of uptake ratio, and thus was not involved in MT921 transport, according to US FDA guidelines [14], whereas OATP1B3, OAT3, NTCP, and ASBT showed greater than two-fold of uptake ratio, and thus could be involved in transport of MT921. The uptake rate of MT921 was higher in 10 µM of MT921 than 100 µM of MT921, indicating that the transporters may get saturated, thus plateauing the uptake (Table 1).

We confirmed the concentration-dependent uptake of MT921 by OATP1B3, OAT3, NTCP, and ASBT and calculated their kinetic values (K_m_, V_max_) in HEK 293-OATP1B3, -OAT3, -NTCP, -ASBT, and mock cells, with increasing MT921 concentration (5–100 µM for OATP1B3, NTCP, and ASBT, 5–300 µM for OAT3) (Figure 1). Kinetic parameters of OATP1B3-mediated MT921 uptake were K_m_ 27.4 µM, V_max_ 94.88 pmol/mg protein/min, intrinsic clearance (V_max_/K_m_) 3.46 µL/mg protein/min, and CL_diffusion_ 1.33 µL/mg protein/min (Figure 1A). In the OAT3-mediated MT921 uptake, K_m_, V_max_, intrinsic clearance, and CL_diffusion_ were calculated as 133.33 µM, 170.76 pmol/mg protein/min, 1.28 µL/mg protein/min, and 1.14 µL/mg protein/min, respectively (Figure 1B). For NTCP, K_m_, V_max_, intrinsic clearance, and CL_diffusion_ were 30.54 µM, 409.28 pmol/mg protein/min, 13.4 µL/mg protein/min, and 1.27 µL/mg protein/min, respectively (Figure 1C). In the case of ASBT, K_m_, V_max_, intrinsic clearance, and CL_diffusion_ were calculated as 34.9 µM, 302.31 pmol/mg protein/min, 8.66 µL/mg protein/min, and 2.03 µL/mg protein/min, respectively (Figure 1D).

#### 2.1.2. Inhibitory Effects of MT921 on The Transport Activities OATP1B1, OATP1B3, OAT3, NTCP, and ASBT

Inhibition of MT921 transport by OATP1B1, OATP1B3, OAT3, NTCP, and ASBT was evaluated using HEK 293-OATP1B1, -OATP1B3, -OAT3, -NTCP, and -ASBT stable cells. The uptake rate was determined by the uptake of probe substrates (0.022 µM [^3^H]estrone-3-sulfate for OATP1B1 and OAT3, 0.06 µM [^3^H]estradiol-17β-D-glucuronide for OATP1B3, 0.8 µM [^3^H]taurocholate for NTCP and ASBT) into transporter-expressing cells and mock cells in the absence and presence of MT921 or representative inhibitors (30 µM cyclosporine A for OATP1B1 and OATP1B3, 50 µM diclofenac for OAT3, 30 µM bromsulphthalein for NTCP, 100 µM deoxycholate for ASBT). The positive inhibitors strongly inhibited the transport of MT921 by OATP1B1, OATP1B3, OAT3, NTCP, and ASBT, and the in vitro test systems were capable of detecting inhibitors of respective transporters. Over the concentration range tested (0.5–100 µM), MT921 showed a concentration-dependent inhibition manner with IC_50_ values of 63.2 µM for OATP1B1, 38.3 µM for OATP1B3, 68.7 µM for OAT3, 53 µM for NTCP, and 39.1 µM for ASBT (Figure 2).

### 2.2. PBPK Modeling and Simulation

#### 2.2.1. MT921 Model

In the final PBPK model, total hepatic clearance, ASBT, NTCP, OAT3, OATP1B3, Glomerular Filtration Rate (GFR), and enterohepatic circulation were utilized to describe the pharmacokinetic properties of MT921. Drug-dependent parameters are summarized in Appendix A. Predicted plasma concentration of the simulated population was compared to plasma concentration-time profile of three doses of MT921 from a clinical trial (mentioned in the method section). Observed data of all doses were appropriately predicted in simulation. Semilogarithmic and linear plots of the simulated population for three dosages are shown in the Appendix A. A MT921 goodness-of-fit plot for the predicted plasma concentrations versus observed plasma concentrations of all studies was made. All observed data were included in 2-fold deviation (Appendix A). The mean relative deviation (MRD) value for all predicted plasma concentration-time profiles was 1.13 (1.08–1.24) and geometric mean fold error (GMFE) values of AUC and C_max_ were 1.10 (1.01–1.17) and 1.09 (1.04–1.16) (Appendix A). All predicted AUC and C_max_ values were within the tw-fold compared with the observed AUC and C_max_. Detailed values of each dose are listed in Appendix A. Sensitivity was analyzed with a simulation of 150 mg MT921. The MT921 model in the 150 mg was sensitive to the properties of the lipophilicity (−2.35), the fraction unbound (−0.71), permeability (−0.07), and total hepatic clearance (−0.05). The result of sensitivity analysis is illustrated in Appendix A.

#### 2.2.2. Amlodipine Model

The final model of amlodipine (AMLO) contains information about hepatic clearance, GFR, and K_i_ values of ASBT. The drug-dependent parameters of AMLO are listed in Appendix A. We compared population simulation to the observed plasma concentration-time profile from 19 clinical studies. All population simulations adequately predicted the results from the clinical studies. Semilogarithmic and linear plots of population simulation of 19 clinical studies are shown in the Appendix A, respectively. The goodness-of-fit of AMLO confirmed that most of the observed values were included within two-fold deviation lines (Appendix A). The MRD of AMLO was 1.29. MRD values for all predicted plasma concentration-time profiles were within the two-fold deviation. GMFE values of AUC and C_max_ were 1.16 and 1.19, repectively, also within the two-fold deviation (Appendix A). Sensitivity analysis results of a simulation of 10 mg AMLO are illustrated in Appendix A. The final model of AMLO was sensitive to the unbound fraction (−0.82), total hepatic clearance (−0.64), lipophilicity (0.46), and permeability (−0.24).

#### 2.2.3. Prediction of Potential DDI between MT921 and Chronic Disease Drugs

With the developed PBPK model of MT921, AMLO, simvastatin (SIMV), and pioglitazone (PIO), potential DDI was tested when these chronic disease drugs were co-administered with MT921. The highest doses of single or multiple drugs were administered once a day for 9 days. At 10 days, 150 mg of MT921 was co-administered with chronic disease drugs. The plasma concentration of MT921 was predicted up to 24 h after administration. In all scenarios, the concentration of MT921 alone (purple line) was within the range of 5–95% concentration of MT921 with chronic disease drugs (pink shade), regardless of which drugs were taken (Figure 3). This means that MT921 concentration did not change with co-administration to AMLO, SIMV, and PIO. Linear and semilogarithmic graphs for comparing only plasma concentration of MT921 alone and MT921 co-administration are shown in Appendix A.

## 3. Discussion

MT921 is a fat-solubilizing drug, injected subcutaneously into the adipose tissue lying anterior to the platysma muscle to reduce submental fat. Since this drug is administered directly into topical sites, its bioavailability is expected to be very low. This was confirmed by a Phase I clinical trial of MT921, where a minimal amount of the drug was found in the systemic circulation of healthy volunteers administered MT921 subcutaneously. Nonetheless, we decided to investigate the in vitro pharmacokinetic DDI potential of MT921 when we had enough of the intended patients, many of whom were expected to be on multiple medications. Furthermore, we could simulate the change of MT921 AUC and C_max_ when patients took drugs for their pre-existing metabolic conditions with MT921. Our in vitro study and in silico PBPK modeling predicted that there would be no potential DDI between MT921 and chronic disease drugs.

Our results revealed that MT921 seemed to have no significant inhibitory effect on cytochrome P450 (CYP) and the UDP-glucuronosyltransferase (UGT) enzyme or inductory effect on CYP isoforms and UGT isoforms (Appendix A) (IC_50_ > 100 μM on CYP isoforms and UGT isoforms; less than two-fold change in the mRNA expression level of CYPs and UGTs compared with vehicle control). We confirmed that MT921 had an extremely rare chance of DDI with the metabolic enzymes recommended by FDA. In addition, MT921 was determined to not be a substrate and did not show inhibitory effects on efflux (MDR1, BCRP, MRP2, and BSEP) or uptake (OATP1B1, OATP2B1, OAT1, OCT1, OCT2, MATE1, and MATE2K) transporters (Table 1, Appendix A, Appendix A). However, OATP1B3, OAT3, NTCP, and ASBT seemed to be involved in MT921 transport (Figure 1 and Figure 2). Therefore, in our MT921 model, CYP and UGT-associated metabolic processes were not considered, but ASBT, NTCP, OAT3, and OATP1B3 were included in our final model to determine the potential drug interaction with MT921 (Figure 1 and Figure 2).

Our PBPK models of MT921 and AMLO describe and predict plasma concentration-time profiles over a dose range, as per the administration protocol from clinical trials published previously [15,16,17,18,19,20,21,22,23,24,25,26,27,28,29]. The performance of these models was demonstrated by the evaluation method. Physiology-based bile acid (PBBA) and the PBPK model of AMLO were developed [30,31,32] to investigate bile recirculation and AMLO DDI prediction with other drugs. Therefore, we first developed a model of CA administration to reduce submental fat and test DDI of CA with AMLO, SIM, and PIO.

In the PBPK model development, transporter K_m_ values obtained from in vitro tests were used to describe pharmacokinetic characteristics, after adjusting the scale used in PK-sim. K_i_ values of the transporter reflected auto-inhibition and DDI and were calculated from the in vitro IC_50_ value. Total hepatic clearance, lipophilicity, unbound fraction, first-order absorption, permeability, and intestinal specific permeability were optimized. Total hepatic clearance of [24-^14^C] CA was included in metabolizing enzymes to cover unknown hepatic clearance [33]. Since the lipophilicity and unbound fraction data were sparse, they were optimized and found to be within the reported range of values proving its accuracy [34,35]. There was no data for absorption or distribution of MT921 after subcutaneou administration. Therefore, first-order absorption, permeability, and intestinal specific permeability were optimized to predict absorption and distribution of MT921. Three dosages of observed data were utilized in the evaluation step. Despite the lack of numbers of clinical trials in evaluation, MRDs and GMFEs of AUC and C_max_ were within two-fold deviation. The GOF and population simulation results also verified correct predictions of observed data using the final PBPK model. Nonlinear PK of MT921 for different doses was also described well. After that, MT921 model sensitivity was analyzed to identify the parameter that could have an impact on the AUC of this model [36]. If the sensitivity value is −1.0, a 10% increase in the parameters leads to a 10% decrease of the PK parameter value. Similarly, if a sensitivity value is 0.5, a 10% increase in the parameters leads to a 5% increase in the PK parameter value. The results indicated that the AUC of MT921 is sensitive to lipophilicity and the unbound fraction, but not K_cat_ and K_i_ of the transporter. It is speculated that MT921 DDI would not be associated with those transporters.

To predict ADME properties of amlodipine, physicochemical properties, Weibull, total hepatic clearance, GFR, permeability, and intestine specific permeability were implemented. Amlodipine is majorly metabolized by CYP3A4 and CYP3A5 [37]. Although it could be more accurate to implement these metabolizing enzymes, total hepatic clearance was utilized as the dominant metabolic process in our model. As Weibull, permeability, and intestinal specific permeability could not be extracted from the published paper, these values were optimized in our model. Amlodipine model was evaluated using 19 clinical trials. Lastly, K_i_ value of ASBT was added to develop the amlodipine model to study DDI. The MRD of amlodipine was 1.29. GMFEs of AUC and C_max_ were 1.16 and 1.19, respectively, and MRD and GMFEs were within two-fold. Sensitivity of the amlodipine model was analyzed to investigate parameters that affect drug exposure. As mentioned above, if the sensitivity value was bigger, the parameter greatly affected the PK parameter. The results show that the amlodipine model was sensitive to unbound fraction, total hepatic clearance, lipophilicity, and permeability. Other parameters decreased drug exposure, while lipophilicity increased drug exposure.

In the SIMV and PIO final model, the reproducibility of the model was first confirmed, and then these models were used. The K_i_ value for transporter was added to inhibit the transporter in DDI simulation [38,39,40,41].

In the simulation step, scenarios of DDI were set, as well as the period in which inhibition could occur. The highest doses of three chronic disease drugs were administered, along with the MT921. K_i_ values of MT921 were obtained from the in vitro test. AMLO, SIMV, and PIO K_i_ values were taken from literature. All of these drugs were assumed to competitively inhibit each other [42]. Before DDI simulation, drug interaction among chronic disease drugs was investigated. DDI between AMLO and SIMV was reported [43]. When SIMV was co-administered with AMLO, AUC and C_max_ of SIMV increased 1.8- and 1.9-fold, respectively. DDI between AMLO and SIMV was reflected in our DDI simulation by adjusting the SIMV final model. In DDI prediction simulation, MT921 concentration with chronic disease drugs was not significantly different from those with MT921 administration alone. In other words, MT921 PK was not affected by transporter inhibition. Since the concentration of MT921 did not change, we checked whether inhibition functions would work well in DDI simulation. Considering the sensitivity value of the MT921 transporter, it is confirmed that the concentration of MT921 changed when the inhibition was changed by the dose or K_i_ value. Therefore, the inhibition of the transporter was well implemented in DDI simulation. These results matched with inhibitor concentration and its IC_50_. C_max_ of AMLO and SIM in the clinic was 2000 times lower than their IC_50_. For PIO, C_max_ of ASBT was 30 times lower and NTCP was two times lower than its IC_50_, though C_max_ of OAT3 was two times higher than its IC_50_. Since MT921 is rarely distributed in the kidney, where OAT3 is located (data not shown), PIO would not inhibit MT921.

Due to the lack of observed data, the MT921 model development and DDI simulation had a limitation. Although the MT921 model showed good results in the model evaluation, results may not be highly reliable because the same data was used for model development and evaluation, due to the lack of clinical trials. Moreover, since DDI simulation had no real-world data, all potential DDI simulations were predicted models, not developed models. These limitations should be considered while interpreting these models.

## 4. Materials and Methods

### 4.1. Materials

The active ingredient of MT921, cholic acid (CA), was synthesized from Medytox Inc. (Suwon, Korea). Cyclosporine A, diclofenac, bromsulphthalein, deoxycholate, chenodeoxycholic acid, and Dulbecco’s Phosphate Buffered Saline (DPBS) were purchased from Sigma-Aldrich (St. Louis, MO, USA). [^3^H]estrone-3-sulfate (45 Ci/mmol), [^3^H]estradiol-17β-d-glucuronide (34.3 Ci/mmol), and [^3^H]taurocholate (5 Ci/mmol) were purchased from Perkin Elmer (Boston, MA, USA). Fetal bovine serum (FBS), non-essential amino acids, penicillin, and streptomycin were purchased from Gibco BRL, Life Technologies (Grand Island, NY, USA). Dulbecco’s Modified Eagle’s Medium (DMEM), poly-d-lysine coated 24-well plates, and poly-d-lysine coated 96-well plates were purchased from Corning-Gentest (Tewksbury, MA, USA). The acetonitrile used was analytical grade and purchased from Merck (Darmstadt, Germany).

The HEK 293-OATP1B1, -OATP1B3, -OAT3, -NTCP, and -ASBT stable cells and mock cells were purchased from Corning Life Science (Woburn, MA, USA).

### 4.2. In Vitro Effect on Transporters

#### 4.2.1. Substrate Specificity of MT921 for OATP1B1, OATP1B3, OAT3, NTCP, and ASBT

The HEK 293-OATP1B1, -OATP1B3, -OAT3, -NTCP, and -ASBT stable cells and mock cells were maintained in DMEM supplemented with 10% FBS, 1% non-essential amino acids, and 100 U/mL of penicillin-streptomycin at 37 °C in a humidified atmosphere of 5% CO_2_ [44]. For the experiments, 0.8 × 10^5^ cells per well were seeded in poly-D-lysine coated 24-well plates, except for HEK293-ASBT cells, where 1 × 10^5^ cells per well were seeded in poly-D-lysine coated 24-well plates. After incubation for 48 h, the medium was removed and the cells were washed with DPBS and preincubated for 10 min in DPBS at 37 °C.

The uptake of MT921 (10 µM and 100 µM) or probe substrates (0.022 µM [^3^H]estrone-3-sulfate for OATP1B1 and OAT3, 0.06 µM [^3^H]estradiol-17β-d-glucuronide for OATP1B3, 0.8 µM [^3^H]taurocholate for NTCP and ASBT) was measured in the absence and presence of representative inhibitors (30 µM cyclosporine A for OATP1B1 and OATP1B3, 50 µM diclofenac for OAT3, 30 µM bromsulphthalein for NTCP, 100 µM deoxycholate for ASBT) [45,46,47,48,49]. After 10 min at 37 °C uptake, the cells were washed twice with ice-cold DPBS. Cells were disintegrated in 150 µL of 70% acetonitrile containing internal standard (IS; 200 nM of chenodeoxycholic acid). The cells were sonicated for 3 s and centrifuged at 13,000 rpm for 10 min. An aliquot (5 µL) from the supernatant (100 µL) was injected into LC-MS/MS system.

Concentration-dependent MT921 uptake by OATP1B3, OAT3, NTCP, and ASBT was examined in HEK 293-OATP1B3, -OAT3, -NTCP, -ASBT and mock cells for 10 min at 37 °C. The uptake rates were measured with increasing MT921 concentration (5–100 µM for OATP1B3, NTCP, and ASBT, 5–300 µM for OAT3). The sample preparation method is as stated above.

#### 4.2.2. Inhibitory Effects of MT921 on the Transport Activities of OATP1B1, OATP1B3, OAT3, NTCP, and ASBT

HEK-293 cells with respective transporters were prepared, as shown in 4.2.1. To examine the inhibitory effects of MT921 on transport activities, the uptake of 0.022 µM [^3^H]estrone-3-sulfate for OATP1B1 and OAT3, 0.06 µM [^3^H]estradiol-17β-d-glucuronide for OATP1B3, and 0.8 µM [^3^H]taurocholate for NTCP and ASBT was measured in the presence of MT921 (0.5–100 µM) [45,46,47,48,49]. After 5 min at 37 °C incubation, the cells were washed twice with ice-cold DPBS. The cells were disintegrated in 0.1 N sodium hydroxide for 1 h. Radioactivity in the samples was measured using a liquid scintillation counter.

#### 4.2.3. LC-MS/MS Analysis

MT921 was analyzed by modifying the protocol from a previously published paper, using an Agilent 6410 Triple Quadrupole LC-MS/MS system (Agilent, Wilmington, DE, USA) equipped with an Agilent 1200 series HPLC system [50]. MT921 was separated using an XBridge C18 column (2.1 mm × 100 mm, 3.5 µm; Waters, Milford, MA, USA). The mobile phases consisted of water and acetonitrile (40:60 *υ*/*υ*) with 0.1% formic acid at a flow rate of 0.2 mL/min. The retention times of MT921 and chenodeoxycholic acid (IS) were 2.1 min and 3.4 min, respectively. Quantitation was carried out using the multiple reaction monitoring (MRM) mode at *m*/*z* 407.5 → 407.5 (collision energy (CE) of 20 eV; negative ion mode) for MT921 and *m*/*z* 391.3 → 391.3 (CE of 25 eV; negative ion mode) for IS. The analytical data was processed by MassHunter software (version B.01.04).

#### 4.2.4. Data Analysis

The uptake of MT921 into HEK 293-OATP1B1, -OATP1B3, -OAT3, -NTCP, and -ASBT stable cells was calculated as percentages relative to that in mock cells. Kinetic parameters of OATP1B3-, OAT3-, NTCP-, and ASBT-mediated MT921 uptake were fit to a modified Michaelis–Menten equation ((ν = (V_max_ × [S])/(K_m_ + [S]) + CL_diffusion_ × [S])) using Phoenix WinNonlin (version 2.1; Pharsight, Mountain view, CA, USA) [51]. V_max_ represents the maximum velocity at saturating substrate concentration, [S] represents the substrate concentration, K_m_ represents the substrate concentration at half V_max_, and CL_diffusion_ represents the passive diffusion clearance.

The degrees of inhibition of transport of OATP1B1, OATP1B3, OAT3, NTCP, and ASBT by MT921 were calculated as percentages of control in the absence and presence of the inhibitors. IC_50_ values were fit to an inhibitory effect equation ((ν = E_max_ × (1 − [I]/(IC_50_ + [I])) using Phoenix WinNonlin (version 2.1; Pharsight, Mountain view, CA, USA) [52]. E_max_ represents the maximum effect, [I] represents the inhibitor concentration, and IC_50_ represents the drug concentration at half inhibition.

### 4.3. PBPK Modeling and Simulation

#### 4.3.1. Software

The PBPK model of MT921 and AMLO were developed using PK-sim^®^ (open systems pharmacology site 9.1 www.open-systems-pharmacology.org (accessed on 21 January 2021)). Model parameter optimization (Monte–Carlo algorithm) and sensitivity analysis were performed using PK-sim^®^. Plasma concentration-time profiles from the literature were digitized with WebPlotDigitizer Version 4.4 [53]. Calculation of quantitative model evaluation, PK parameter analysis, and graph plotting were accomplished with R 4.0.2 (the R foundation for statistical computing) and R studio 1.4.1103 (R studio, Inc, Boston, MA, USA).

#### 4.3.2. PBPK Model Development

The PBPK model was developed using in vitro, in vivo, and clinical study data. Information of physicochemical properties, absorption, distribution, metabolism, and excretion (ADME) were used to reproduce compound characteristics. Clinical studies data (observed data) were used to create a data set, consisting of a training set and test set, for model development and evaluation. Demographic data of healthy individuals was used. If a clinical study did not provide demographic information, default values from the PK sim population database were used. Metabolic enzymes and transporter proteins were implemented using the PK-sim gene expression database [54]. Model parameters were then optimized by PK sim calculation methods for fitting predicted simulation to observed concentration of clinical study.

#### 4.3.3. MT921 Model Development

MT921 in vitro data about the transporter and a clinical study containing demographic information (age, height, and weight) were provided by Medytox Inc. The rest of the in vitro data about physicochemical properties and ADME information of MT921 were retrieved from published literature. To develop a MT921 model, physicochemical properties, ASBT, NTCP, OAT3, OATP1B3, total hepatic clearance, GFR, and EHC recirculation was implemented. We found that MT921 is a substrate of ASBT, NTCP, OAT3, and OATP1B3, and MT921 can inhibit ABST, NTCP, OAT3, and OATP1B3. Experimental K_m_ and V_max_ values of ASBT, NTCP, OAT3, and OATP1B3 were utilized. The K_i_ value was calculated from the IC_50_ value using the Cheng–Prusoff equation
(1)$$K_{i}=\frac{{\mathrm{IC}}_{50}}{1+\frac{\left[S\right]}{K_{m}}}$$
where K_i_ is the inhibition constant, IC_50_ is half of the maximal inhibitory concentration of MT921, [S] is the concentration of substrate, and K_m_ is the substrate concentration required for half of the maximum rate of transport.

These values are shown in Figure 1 and Figure 2. K_cat_ of MT921 was calculated by the PK-sim-embedded Michaelis-Menten calculation method. To explain unknown clearance, total hepatic clearance was used. Total hepatic clearance was obtained from [24-^14^C] CA clearance [33]. GFR and EHC recirculation values were assumed to be 1.

One clinical study was used as a training set; MT921 of 60 mg, 120 mg, and 150 mg were administered subcutaneously. If there was no information on sex in the clinical study, the population was assumed to be 100% male. A list of clinical studies is shown in Appendix A. Partition coefficients and cellular permeability were taken from those calculated by Schmitt [55] and the PK-sim standard calculation method. Model parameters that could not obtain exact values from literature were optimized to observe the data of the training set.

#### 4.3.4. Amlodipine Model Development

To develop the AMLO model, data on physicochemical properties, information about ADME, and clinical studies of AMLO were extracted from published literature. Total hepatic clearance and GFR were implemented to describe metabolism and excretion. Among the 19 clinical studies with two repeated doses and 17 single doses, 7 clinical studies are used as the training set and 12 as the test set. All AMLO was administered orally, 2.5–10 mg. Asian [56] demographic information was used for Korean and Chinese subjects whose demographic data was not provided. European [57] and Japanese (2015) demographic information was used for Caucasian and Japanese. All 19 clinical studies are shown in Appendix A. Partition coefficients and cellular permeability was calculated using the Rogers and Rowland method [58,59] and PK-sim standard method. Model parameters whose exact values were not obtained from literature were optimized to fit the predicted simulation to observed data.

#### 4.3.5. PBPK Model Evaluation

For model evaluation, several methods were used. As a visual comparison of the model performance, population simulations and goodness-of-fit were conducted. In total, 100 individuals were used in population simulations. If no demographic information was available, 20–50 years of age and male gender were assumed. The population predicted plasma concentration-time profile was compared to that observed in clinical data. Comparison of predicted versus observed plasma concentration of all studies was plotted in the goodness-of-fit plot. As a quantitative measure of performance of the model, the MRD of all PK parameters and the GMFE of all predicted PK parameters were calculated. If MRD and GMFE values were less than 2, it was considered to be adequate model performance.

The equation of MRD and GMFE are:(2)$$\mathrm{MRD}=10^{x},\;x=\sqrt{\frac{\sum_{i=1}^{n}{\left(\log_{10}C_{\mathrm{predicted},\;i}-\log_{10}C_{\mathrm{observed},\;i}\right)}^{2}}{n}}$$
where C_predicted,i_ is predicted plasma concentration, C_observed,i_ is observed data from literature, and n is the number of observed values.
(3)$$\mathrm{GMFE}=10^{x},\;x=\frac{\sum_{i=1}^{m}\left|\log_{10}\frac{{\mathrm{PK}\;\mathrm{parameter}}_{\mathrm{predicted},i}}{{\mathrm{PK}\;\mathrm{parameter}}_{\mathrm{observed},\;i}}\right|}{m}$$
where PK parameter_predicted,i_ is predicted AUC and C_max_, PK parameter_observed,i_ is observed AUC and C_max_ from literature, and m is the number of studies.

#### 4.3.6. Sensitivity Analysis

The sensitivity analysis of the final PBPK model was performed to determine how other parameters affect PK parameters. All parameters that could influence PK were included. Sensitivity analysis was performed for the simulation using the highest dose of the drug (MT921 150 mg, AMLO 10 mg) and conducted with the Sensitivity Analysis tool in PK-sim. The variation range and maximum number of steps were set to 10 and 9, respectively. The sensitivity of each parameter was calculated as the ratio of the relative change in the AUC to the relative change in the parameter.

The equation is below:(4)$$S=\frac{\Delta \mathrm{AUC}}{\Delta P}\times \frac{P}{\mathrm{AUC}}$$
where S is the sensitivity of the area under the curve; AUC is the area under the curve; ΔAUC is the change in the area under the curve; ΔP is the change in the model parameter value; P is the original parameter value in final model.

#### 4.3.7. Prediction of Potential DDI between MT921 and the Inhibitors of Uptake Transporters

MT921 can be co-administered with SIMV, AMLO, and PIO. These drugs can affect MT921 transport. SIMV inhibits ASBT and NTCP [38,39], and AMLO inhibits ASBT [60]. PIO inhibits ASBT, NTCP, and OAT3 [38,40,41] (Figure 4).

To predict the potential DDI of MT921, SIMV and PIO models already developed by Hanke, along with MT921 and AMLO PBPK models, were used [61,62]. K_i_ values of ASBT, NTCP, OAT3, and OATP1B3 obtained from in vitro tests and literature were added to developed PBPK models. Inhibition of ASBT (K_i_ = 54.60 µM) [38], NTCP (K_i_ = 4.04 µM) [40], and OAT3 (K_i_ =1.02 µM) [41] was implemented by PIO. Inhibition of ASBT (K_i_ =10.40 µM) [38] and NTCP (K_i_ = 47.90 µM) [39] was implemented by SIMV. Inhibition of ASBT (K_i_ = 42.10 µM) [61] was implemented by AMLO. In the simulation for investigating potential DDI, the highest dose of AMLO, PIO, and SIMV was administered once a day for 10 days based on each scenario. At 10 days, MT921 150 mg was administered subcutaneously. Potential DDI was predicted with single or multiple drugs. The scenario simulation is presented in Figure 5.

To estimate changes in PK parameter of MT921, DDI PK parameter ratio was calculated using PK parameters of MT921 administered alone and co-administered.

The equation of PK parameter ratio is below:(5)$$\mathrm{DDI}\;\mathrm{PK}\;\mathrm{parameter}\;\mathrm{ratio}=\;\frac{{\mathrm{PK}\;\mathrm{parameter}}_{\;\mathrm{MT}921\;\mathrm{during}\;\mathrm{co}-\mathrm{administration}}}{{\mathrm{PK}\;\mathrm{parameter}}_{\mathrm{MT}921\;\mathrm{alone}\;}}$$
where PK parameter is AUC and C_max_.

## 5. Conclusions

To verify the DDI of MT921s with other drugs, we conducted various in vitro assays according to the DDI guidelines by the FDA, as well as in silico prediction using PBPK models. In summary, none of the results of in vitro assays indicated a significant DDI risk for metabolism or transport, considering the exposure from phase 1 clinical trials of MT921. Furthermore, we found no DDI possibility of MT921 with potential co-administrated drugs related to metabolic disease syndrome, using the PBPK modeling approach in various scenarios. Therefore, MT921 could be administered without a DDI risk based on in vitro study and related in silico simulation. Further clinical studies are needed to validate this finding.

## Figures and Tables

**Figure 1 pharmaceuticals-14-00654-f001:**
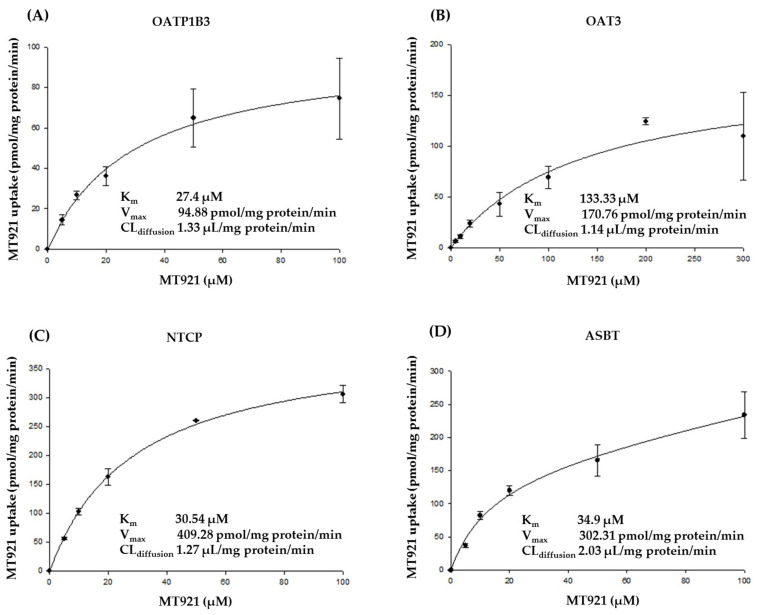
Concentration-dependent uptake of MT921 (5–100 µM for OATP1B3, NTCP, and ASBT, 5–300 µM for OAT3) in HEK 293 (**A**) -OATP1B3; (**B**) -OAT3; (**C**) -NTCP; (**D**) -ASBT. Each bar represents mean ± standard deviation from triplicate experiments. V_max_, maximum velocity at saturating substrate concentration; [S], substrate concentration; K_m_, substrate concentration at half of V_max_; CL_diffusion_ represents the passive diffusion clearance.

**Figure 2 pharmaceuticals-14-00654-f002:**
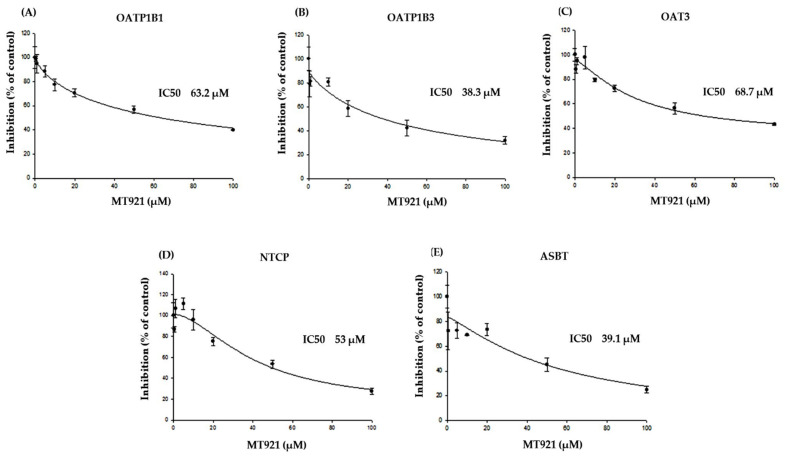
Concentration-dependent inhibition of MT921 uptake (0.5–100 µM) on (**A**) 0.022 µM [^3^H]estrone-3-sulfate in HEK 293-OATP1B1; (**B**) 0.06 µM [^3^H]estradiol-17β-D-glucuronide in HEK 293-OATP1B3; (**C**) 0.022 µM [^3^H]estrone-3-sulfate in HEK 293-OAT3; (**D**) 0.8 µM [^3^H]taurocholate in HEK 293-NTCP; (**E**) 0.8 µM [^3^H]taurocholate in HEK 293-ASBT. Each bar represents mean ± standard deviation from triplicate experiments. IC_50,_ drug concentration at 50% inhibition.

**Figure 3 pharmaceuticals-14-00654-f003:**
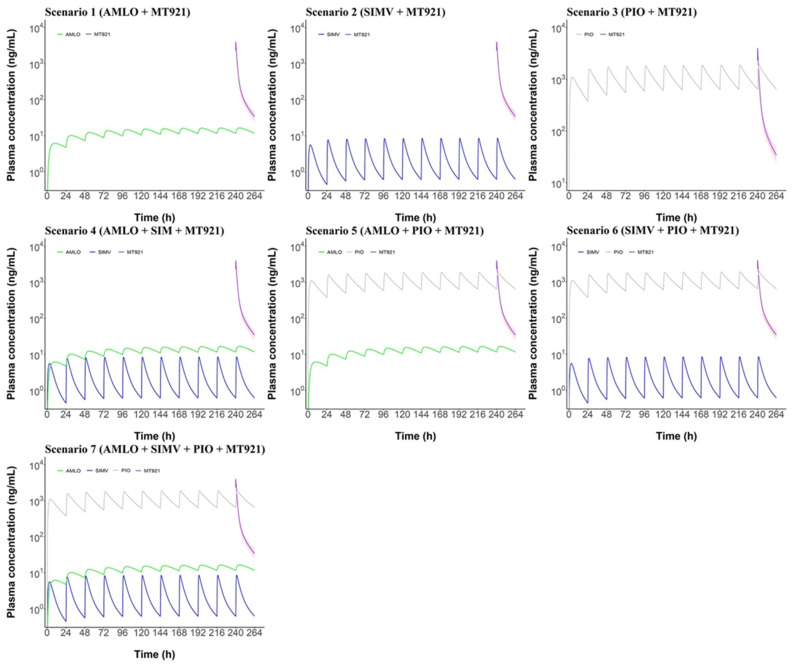
Semilogarithmic graph comparing plasma concentration of MT921 administered alone and co-administered. Chronic disease drugs were administered for 9 days, once a day. At 10 days, 150 mg of MT921 was co-administered with chronic disease drugs. After administration, MT921 concentration was predicted. The purple line is a MT921-only population plasma concentration. The pink shade area is the 5–95% range of MT921 co-administration population plasma concentration. The green line is AMLO. The blue line is SIM. The gray line is PIO.

**Figure 4 pharmaceuticals-14-00654-f004:**
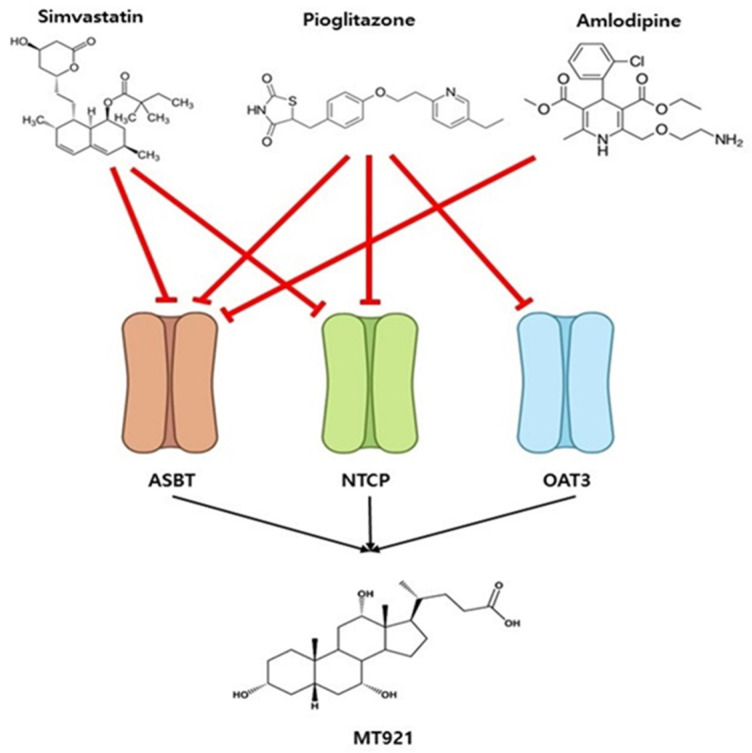
Drug–drug interaction networks. Simvastatin, amlodipine, and pioglitazone are the perpetrator drug in the DDI prediction model. MT921 (Cholic acid) is the victim drug. Simvastatin inhibits ASBT and NTCP. Amlodipine inhibits ASBT. Pioglitazone inhibits ASBT, NTCP, and OAT3. The red solid line represents inhibition, and the black solid line represents transport.

**Figure 5 pharmaceuticals-14-00654-f005:**
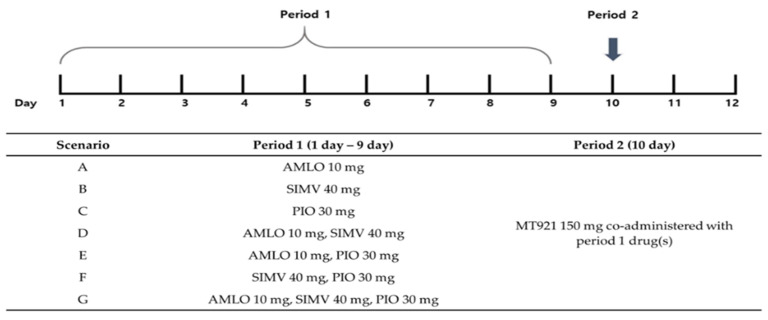
DDI scenario. During period 1, DDI drug(s) was administered as q.d., and MT921 was co-administered with DDI drug(s). AMLO, amlodipine; SIMV, simvastatin; PIO, pioglitazone.

**Table 1 pharmaceuticals-14-00654-t001:** The uptake transport rate of MT921 (10 µM and 100 µM) in HEK 293-OATP1B1, -OATP1B3, -OAT3, -NTCP, -ASBT, and mock cells.

Concentration (µM)	Transporter	Uptake (pmol/mg protein/min)	Uptake Ratio (Transporter vs. Mock)
10	Mock	7.61 ± 0.91	1.00
	HEK 293-OATP1B1	12.68 ± 0.64	1.67
100	Mock	72.15 ± 3.91	1.00
	HEK 293-OATP1B1	72.98 ± 4.46	1.01
10	Mock	7.61 ± 0.91	1.00
	HEK 293-OATP1B3	23.24 ± 1.08	3.05
100	Mock	72.15 ± 3.91	1.00
	HEK 293-OATP1B3	126.56 ± 5.44	1.75
10	Mock	7.61 ± 0.91	1.00
	HEK 293-OAT3	15.46 ± 2.09	2.03
100	Mock	72.15 ± 3.91	1.00
	HEK 293-OAT3	63.53 ± 5.35	0.88
10	Mock	4.85 ± 0.55	1.00
	HEK 293-NTCP	56.74 ± 6.66	11.71
100	Mock	57.16 ± 0.67	1.00
	HEK 293-NTCP	215.08 ± 8.65	3.76
10	Mock	4.38 ± 0.42	1.00
	HEK 293-ASBT	78.33 ± 5.66	17.89
100	Mock	50.22 ± 5.65	1.00
	HEK 293-ASBT	253.49 ± 13.66	5.05

Each data represents the mean ± standard deviation from triplicate determinations.

## Data Availability

Data is contained within the article and Appendix A.

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
