# Peer review of "Evaluation for Potential Drug–Drug Interaction of MT921 Using In Vitro Studies and Physiologically–Based Pharmacokinetic Models"

_pharmaceuticals, 2021, doi:10.3390/ph14070654_

Round 1

Reviewer 1 Report

The manuscript discussed the transporter characteristics of MT921 and its inhibition potential for drug-metabolizing enzymes and transporters. A thorough in vitro perpetrator inhibition package is generated.  However, there is not a lot of details on victim drug-drug interaction potential.  The specific comments are:

  1. MT921 is a substrate for a number of transporters. What is the clearance mechanism of M921?  What are the contributions of renal, biliary and metabolism?  How many is reabsorbed? These data are important to predict victim DDI.
  2. What is the fractional clearance from each of the pathways, i.e., hepatic, renal? What transporters are involved in each of the mechanisms, i.e., what is the fractional transport?
  3. Are there any enzymes involved in the metabolism of the compound?
  4. How does OATP inhibitor affect MT921 exposure?
  5. Have the simvastatin, pioglitazone and amlodipine PBPK files been validated with all available clinical data?

Author Response

Reviewers' comments

  • We thank the reviewers for reviewing our manuscript and providing valuable comments, which have substantially improved our manuscript. Our responses to the reviewer’s comments, together with the manuscript’s alterations, are as follow:

Reviewer 1

Comments to the Author

: The manuscript discussed the transporter characteristics of MT921 and its inhibition potential for drug-metabolizing enzymes and transporters. A thorough in vitro perpetrator inhibition package is generated. However, there is not a lot of details on victim drug-drug interaction potential. The specific comments are:

  1. MT921 is a substrate for a number of transporters. What is the clearance mechanism of MT921? What are the contributions of renal, biliary and metabolism? How many is reabsorbed? These data are important to predict victim DDI.
  • We thank you for reviewer’s comment. Cholic acid (CA), the active ingredient in MT921, is one of the primary bile acids (BA) endogenously produced in the liver. The BAs are conjugated to taurine or glycine in the liver mainly via two enzymes, BA CoA synthase, and BA-CoA-amino acid N-acetyltransferase, and excreted through the canaliculi to the biliary system. Furthermore, in the small and large intestine, the bacterial deconjugation, dehydrogenation, 7 alpha-dehydroxylation, and epimerization of the primary BA (cholic acid) produces the secondary BA (deoxycholic acid)*. Because the BA is bio-transformed from primary BA to secondary BA to undergo enterohepatic circulation, the CA as primary BA is not directly reabsorbed in the form of the parent drug. In the references using radio-labeled CA, no evidence for enterohepatic cycling of 14C-cholic acid was found in human**. Furthermore, renal loss of injected 14C did not significantly contribute to the overall clearance since no subject excreted more than 1% of the injected dose during the first 2h after injection.
  • * Di Ciaula, A.; Garruti, G.; Lunardi Baccetto, R.; Molina-Molina, E.; Bonfrate, L.; Wang, D.Q.-H.; Portincasa, P. Bile Acid Physiology. Hepatol. 2017, 16, S4–S14, doi:10.5604/01.3001.0010.5493.
  • ** Engelking, L.R.; Barnes, S.; Dasher, C.A.; Naftel, D.C.; Hirschowitz, B.I. Radiolabelled Bile Acid Clearance in Control Subjects and Patients with Liver Disease. Sci. 1979, 57, 499–508, doi:10.1042/cs0570499.

  1. What is the fractional clearance from each of the pathways, i.e., hepatic, renal? What transporters are involved in each of the mechanisms, i.e., what is the fractional transport?
  • We thank you for your comment. The mean hepatic extraction efficiencies are 60-86% in healthy subjects*. As mentioned above, renal clearance is negligible. MT921 (cholic acid) is excreted by renal (0.1%) and feces (56.6%) in rat during 194h after administration (data not shown). Bile acid enters the liver by NTCP, OATP1B3. ASBT and OAT3 are involved in bile acid uptake in the intestine and kidney.
  • * Engelking, L.R.; Barnes, S.; Dasher, C.A.; Naftel, D.C.; Hirschowitz, B.I. Radiolabelled Bile Acid Clearance in Control Subjects and Patients with Liver Disease. Sci. 1979, 57, 499–508, doi:10.1042/cs0570499.

  1. Are there any enzymes involved in the metabolism of the compound?
  • We thank you for your comment. As shown in our result, MT921 has no significant inhibitory effect with cytochrome P450 (CYP), UDP-glucuronosyltransferase (UGT) enzyme, and inductive effect with CYP isoforms and UGT isoforms (Table S1, S2, S6). We confirmed that MT921 had no chance of DDI with the metabolic enzymes recommended by FDA. These results were described in Discussion of the manuscript (revised version page 8, lines 192-196).

  1. How does OATP inhibitor affect MT921 exposure?
  • We thank you for your comment. As shown in our result (Figure 1), OATP1B3 showed greater than 2-fold of uptake ratio, and thus, could be involved in transport of MT921. Therefore, if the OATP inhibitor is coadministered with MT921, uptake and exposure of MT921 could be changed. With assumption it, we simulated several scenarios, including OATP inhibitor administration with MT921.

  1. Have the simvastatin, pioglitazone and amlodipine PBPK files been validated with all available clinical data?
  • We thank you for your comment. The PBPK models of simvastatin, pioglitazone, and amlodipine were validated with available clinical data from the literatures. Clinical data of amlodipine are in the manuscript (page 8, lines 203-205). Clinical data of simvastatin and pioglitazone can be available in published literatures (revised version page 13, lines 448-449).

Reviewer 2 Report

Comments:

    In this manuscript, the uptake of MT921 and the effects of MT921 on the functions of cytochrome P450 (CYP), UDP-glucuronosyltransferase (UGT), and transporters have been determined. The drug interactions of MT921 with amlodipine, simvastatin, and pioglitazone were assessed. Results of the assessments suggest that the interactions of MT921 with these drugs are predicted to be little. Comments are as follows.

  1. On page 6, line 168, Figure 6 should be Figure 5. There was no Figure 6.
  2. On page 8, line 196, please change “show inhibition effect” to “show inhibitory effects”.
  3. Please combine Table 2 and Figure 5 to be one table or figure.
  4. Please provide the pharmacokinetic parameters of MT921 in the study of “Prediction of Potential DDI” (section 2.2.3). To be consistent with the statement on page 13, lines 437-439, it is suggested that the title of this section can be changed to “Prediction of Potential DDI between MT921 and the inhibitors of uptake transporters”.
  5. What did “A 100 individuals were used in individual simulations.” mean? (page 12, lines 410-411) Why was "A" used? Was the IRB approval for human study available?

Reviewer 3 Report

The authors conducted an in vitro DDI study on MT921 using well-established scientific methods. In addition, authors used data from other studies to predict the potential for DDI using the physiologically based pharmacokinetic modeling and simulation. The authors report the absence of significant enzyme/transporter inhibition. The prediction model also found only negligible magnitude of DDI. The authors concluded that "MT921 could be used safely without the concerning DDI problems"

Major comments

  1. On the in vitro incubation studies, only the transporter studies (method and results) are presented in the body of the manuscript. Details of the studies on drug metabolizing enzymes are presented as supplemental files. The manuscript will benefit from the addition of a summarized method/result of the effects of MT921 on the DMEs. More details can remain as supplemental files. But for a complete picture to a reader, the transporter study report should be complemented with DME inhibitory report in the main body of the manuscript.
  2. The conclusion of the authors is an overstatement. Only clinical studies can conclude that a drug "could be used safely without the concerning DDI problems". Less so for an in vitro study that did not consider the potential of MT921 to induce CYP3A4, 2C9, 2C19 etc. Conclusions should reflect the study.

Otherwise, the study is thorough and scientifically sound.

Minor comments

  1. line 29 - 'developedand' - no space
  2. line 293 - Replace 'potential' with 'effect'
  3. line 304 - remove 'were'
  4. line 310 - was 100 microL the injection volume?
  5. line 317 - inser 'of' between 'Activities' and 'OATP1B1'
  6. line 323 - replace 'uptake' with 'incubation'
  7. line 325 - insert 'using' between 'measured' and 'liquid'
  8. line 342 - Phoenix not Pheonix. Also line 349 and any other place
  9. line 358 - Put a period/full stop after '[53]'

Round 2

Reviewer 1 Report

The authors have revised the manuscript.  The reviewer dose not have any additional comments.

Author Response

We checked the 2nd review report.